# ARTIFACTLINKER: LINKING SCIENTIFIC ARTIFACTS FOR AUTOMATIC SOTA DISCOVERY

## ABSTRACT

Scientific artifacts, such as models and benchmarks, are the foundation of machine learning research. With the rapid growth of repositories like HuggingFace, researchers now have access to millions of high-quality artifacts contributed by different researchers, yet the challenge remains: how can we automatically discover the state-of-the-art (SOTA) model for a given benchmark, fully leveraging existing scientific artifacts? We address this task, abbreviated as **automatic SOTA discovery**, by first modeling HuggingFace as an artifact graph, where nodes represent models or benchmarks and edges capture their relationships, labeled with evaluation results. Within this graph, we formulate the automatic SOTA discovery as the process of identifying new unobserved links with high potential performance that could advance future research. To enable scalable and efficient discovery of SOTA artifact links, we propose ARTIFACTLINKER, a two-stage framework for automatic SOTA discovery: (1) prediction, which identifies promising links with Graph Neural Networks (GNNs) or graph-augmented LLMs, and (2) verification, which validates promising predicted links through reproducible and automatic coding experiments and agents. To evaluate ARTIFACTLINKER, we further propose ARTIFACTBENCH, collecting 1,372 models and 308 benchmarks for systematically measuring prediction and verification performance and helping to develop new SOTA discovery agents. Our key results indicate that the graph-based prediction module in ARTIFACTLINKER is effective in prediction. Moreover, an automatic verification pipeline in ARTIFACTLINKER can verify that the identified promising links indeed achieve high performance on existing benchmarks in a fully automatic way.

## 1 INTRODUCTION

Scientific artifacts are the fundamental building blocks of research (Heumüller et al., 2020; Cooper et al., 2022; Johnson et al., 2019). Codebases on the GitHub platform, papers available on arXiv, models and benchmarks on the HuggingFace are all examples of such artifacts. Researchers engaged in doing reproducible and high-quality research share, interact with, and build upon these artifacts, releasing new versions to demonstrate progress (Marić et al., 2023; Lissa et al., 2020). In the machine learning community, a vast number of artifacts (>1M on HuggingFace) are produced by researchers working in different domains (Castaño et al., 2024; Ait et al., 2023; Laufer et al., 2025). This naturally raises an important question: *How can we leverage existing artifacts to enable automatic discovery?* Addressing this question would (1) allow us to utilize diverse types of artifacts better, and (2) promote scalable and automated scientific discovery based on existing resources. We focus on the HuggingFace community as a case study, since it is one of the largest and most active hubs of open-source machine learning artifacts and targeting at making experiments more accessible and easy to run. With countless models, benchmarks, and libraries hosted on the platform, it provides an invaluable foundation for exploring automated discovery.

Building an automatic discovery system on HuggingFace presents several challenges. First, the concept of "automatic discovery" itself is ill-defined (Beel et al., 2025; Kitano, 2021; Kramer et al., 2023)—what does it really mean for a system to conduct research autonomously and contribute to future applications? Second, although HuggingFace provides convenient access to models and datasets, building a fully automated and reproducible pipeline to verify predicted model performance is still difficult (Urbanowicz et al., 2022). The usability of models and benchmarks remains

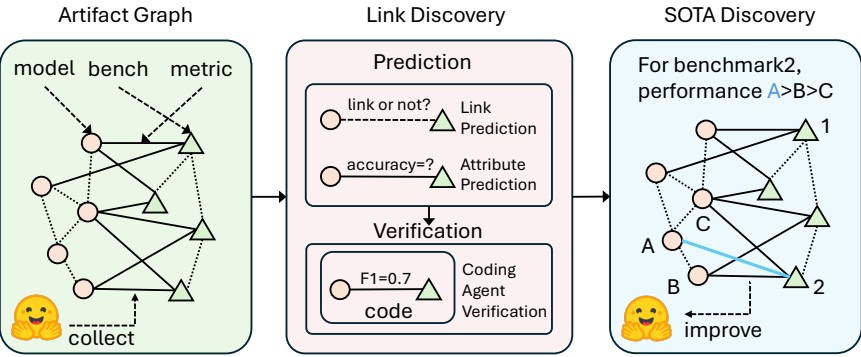

Figure 1: **Overview of ARTIFACTLINKER.** The overall automatic SOTA discovery process in ARTIFACTLINKER includes three main components: (1) artifact graph construction from HuggingFace (models, benchmarks, evaluation relationships); (2) link prediction and verification as discovery tasks; (3) state-of-the-art result filtering. Noticable, besides the model-benchmark edges, model-model edges and benchmark-benchmark edges represent the internal dependency between models and datasets. For example, `arabic-sentiment-bert-model` is a dependence with `bert-base-uncased`. `gsm8k-distilled` is a variant version of the original `gsm8k`. These relations exist conceptually but are not included in our bipartite graph model.

inconsistent, and many of them require specialized configurations to work properly and reproduce research works built on them, even a frustrating process for human researchers (Mu et al., 2025).

**Thinking HuggingFace as an artifact graph.** Our key observation is that the Hugging Face community can be naturally represented as a graph (Chen et al., 2025; Laufer et al., 2025) as shown in Figure 1 (left), where models and benchmarks serve as nodes and their relationships form the edges. This perspective is motivated by three characteristics of the platform: (1) it hosts a large collection of artifacts, including both models and benchmarks; (2) many of these artifacts are high-quality and widely adopted by researchers; and (3) it encodes rich relational information—for example, a model trained on one dataset and evaluated on another with a reported F1 score. Such structured relationships are often difficult to extract directly from research papers. Prior work has largely treated Hugging Face as an information source for retrieval (Silva et al., 2025) or as an API hub (Shen et al., 2023). In contrast, we highlight its value for dynamic discovery. Rather than viewing it as a static data source, we aim to actively uncover new relationships—particularly model–benchmark interactions defined through evaluation metrics. These interactions naturally form the edges of an artifact graph.

**Linking artifacts as automatic SOTA discovery.** Within this artifact graph, we define the task of automatic discovery as finding links with state-of-the-art evaluation results between artifacts (models and benchmarks). Concretely, our goal is to build an auto-discovery engine that leverages the existing artifact graph structures in the Hugging Face community and their existing relationships to propose unfounded connections. In this way, automatic discovery becomes a well-defined problem: identifying state-of-the-art links—such as models that could perform well on unexplored benchmarks—thereby enabling systematic and scalable research progress.

**Novel framework for automatic discovery.** To operationalize the linking problem as automatic discovery, we propose a novel framework with two stages (see Figure 1 (center)): (1) prediction and (2) verification. Given the vast number of artifacts in the graph and the high cost of verifying evaluation results, directly identifying the small subset of valuable links through execution alone is prohibitively costly. Our framework addresses this challenge by first performing prediction, where we use strong priors to filter out the majority of unlikely links—analogous to how experienced researchers prioritize promising directions. In the verification stage, we employ a coding agent to test and validate the predicted links, grounding hypotheses into real, reproducible results. This division of labor enables scalable automatic discovery while ensuring that results remain empirically verifiable.

**Main Discovery.** Our work makes three key contributions: (1) we construct ARTIFACTBENCH, a benchmark for prediction, verification, and automatic discovery, (2) we design a two-stage frame-

work named ARTIFACTLINKER for scalable link discovery, and (3) we demonstrate the practical value of this system through a case study for potential link discovery.

## 2 RELATED WORK

**HuggingFace platform utilization**. HuggingFace has increasingly become a natural platform for studying automatic discovery. Prior works have largely relied on static analyses of its artifacts and relationships to characterize trends in machine learning development (Chen et al., 2025; Laufer et al., 2025). Beyond serving as a repository, HuggingFace has been conceptualized in multiple ways: as a knowledge graph (Silva et al., 2025), an API hub (Shen et al., 2023), a model card aggregator (Yang et al., 2024), and even an evolutionary tree (Gao & Gao, 2023). Other studies have examined its community dynamics (Rahman et al., 2025; Castaño et al., 2023). In contrast, our work moves beyond static description and trend analysis, which are the traditional focus of data mining. Recent efforts have begun to bridge this gap: Liu et al. structures benchmarks into knowledge graphs to assist LLMs in design, while ADGym (Jiang et al., 2023) systematizes design choices for deep anomaly detection. Similarly, we leverage the intrinsic graph structure of HuggingFace artifacts—much like the structured learnware dock system of Beimingwu (Tan et al., 2024)—but focus on grounded, execution-based prediction and verification. This perspective highlights HuggingFace not only as a repository of information but also as a foundation for building an automatic discovery engine—one that functions as a research scaffold.

**Large-scale prediction for accelerating discoveries**. Accelerating scientific discovery has been a major focus in domains such as drug discovery (Stokes et al., 2020; Serrano et al., 2024; Vișan & Negut, 2024; You et al., 2022), materials science (Xie & Grossman, 2018; Butler et al., 2018), and molecular design (Segler et al., 2018), among others (Cheng et al., 2025). In these settings, experimental verification is prohibitively costly and time-consuming, which makes reliable prediction an indispensable first step. By narrowing down the vast search space to a manageable set of promising candidates, prediction enables researchers to prioritize what is worth validating in the lab. Without this predictive stage, discovery in such domains would be prohibitively inefficient. Yet, these tasks remain highly flexible and difficult to evaluate systematically, often requiring research agents that can manage complex, multi-step workflows. In contrast, our work focuses on a more tractable class of automatic discovery tasks. By leveraging the intrinsic linking structure of HuggingFace artifacts, we perform prediction in a structured research ecosystem with a large number of artifacts. This design preserves the benefits of predictive discovery—guiding researchers toward promising directions—while avoiding the prohibitive costs of physical experimentation.

**LLM-based coding agents for reproducible experimentation**. Beyond prediction, an important characteristic of machine learning research is its relatively high reproducibility (Bajwa, 2021). Unlike domains such as biology or chemistry, where experimental verification is costly and time-consuming, verification in machine learning is largely coding-based and thus far more accessible. This makes it feasible to design coding execution tasks that reproduce experiments at scale. Prior work has explored free-form discovery with generating executable code from research ideas (Lu et al., 2024; Jansen et al., 2024; 2025), though evaluating the quality of such code remains difficult because the underlying ideas or intents are often ambiguous. Other efforts have focused on reproducing experiments within specific codebases (Bogin et al., 2024; Starace et al., 2025; Kim et al., 2025; Seo et al., 2025; Siegel et al., 2024; Xiang et al., 2025), a setting that is challenging and costly due to the complexity and idiosyncrasies of large software systems. In contrast, our task narrows the scope: we execute experiments directly by combining a HuggingFace model artifact with a HuggingFace benchmark artifact under a specified evaluation metric. This design preserves the reproducibility benefits of coding execution while avoiding the complexity of open-ended idea generation or large codebase replication.

## 3 BUILDING AN ARTIFACT GRAPH FROM HUGGINGFACE

As the first stage of ARTIFACTLINKER, we first formally provide the definition of artifact graphs that we conduct link discovery on. Furthermore, we provide details about how we extract the artifact graph from the HuggingFace platform.

**Definition of artifact graphs**. We define the artifact graph as an undirected heterogeneous bipartite graph $\mathcal{G} = (\mathcal{V}, \mathcal{E})$, where the node set $\mathcal{V} = \mathcal{V}_m \cup \mathcal{V}_d$ consists of two disjoint types of nodes: **model nodes** $\mathcal{V}_m$ and **benchmark nodes** $\mathcal{V}_d$. The edge set $\mathcal{E} \subseteq \mathcal{V}_m \times \mathcal{V}_d \times \mathcal{K}$ encodes **evaluation relationships**, where each edge $(u, v, k)$ indicates that a model $u \in \mathcal{V}_m$ has an evaluation result on a benchmark $v \in \mathcal{V}_d$ with score $k = \phi(u, v)$ defined by a metric function $\phi : \mathcal{V}_m \times \mathcal{V}_d \to \mathcal{K}$ (e.g., accuracy or F-score). Besides edges, each benchmark node is associated with attributes such as metadata (e.g., download counts) and task descriptions, while each model node is associated with metadata and specifications, including architecture, number of parameters, and configuration. Such rich information on both edges and nodes allows the graph to capture fine-grained performance relationships between models and benchmarks.

**Data collection**. We construct edges in the artifact graph by extracting ground-truth evaluation metrics from Hugging Face model cards[1]. Specifically, we parse the README files to identify model–dataset pairs along with their reported performance scores. Model and dataset names are then matched to their canonical entries on the Hugging Face platform to ensure consistency, yielding a clean set of evaluation edges. In total, this process produces $|\mathcal{E}| = 2{,}067$ perfectly matched evaluation records, which serve as edge attributes in the graph. For nodes, we collect both models and benchmarks. Each model is described by its model card, dataset card, and metadata fields (e.g., architecture, parameters, tags), while each dataset includes metadata such as domain, size, and license. To ensure relevance, we retain only nodes that participate in at least one evaluation edge, filtering out isolated artifacts.

**Graph statistics**. The final artifact graph contains $|\mathcal{V}| = 1{,}680$ nodes, consisting of $|\mathcal{V}_m| = 1{,}372$ models and $|\mathcal{V}_d| = 308$ benchmarks, with $|\mathcal{E}| = 2{,}067$ edges in total. The average degree of a model node is $\bar{d}_m = \frac{|\mathcal{E}|}{|\mathcal{V}_m|} \approx 1.51$, while that of a benchmark node is $\bar{d}_d = \frac{|\mathcal{E}|}{|\mathcal{V}_d|} \approx 6.71$, indicating a sparse bipartite structure. To quantify the community structure, we apply Louvain community detection (Blondel et al., 2008) and compute the modularity, obtaining $Q = 0.9233$, which highlights the presence of clear sub-community structures (e.g., NLP, audio, computer vision) under sparse connectivity. Figure 4 shows the visualization of the artifact graph structure. While benchmarks like ImageNet and MMLU form expected hubs, edge concentration is limited. The average dataset degree is 6.71, and the median is 2.00, indicating broadly distributed connectivity. The top 5 datasets contribute only 25.88% of edges, and 45.13% of datasets have degree $\geq 3$, showing that many non-famous datasets participate in multiple links. Thus, the graph is sparse but not dominated by a few popular nodes, and learning is not restricted to high-degree hubs.

# 4 LINKING SCIENTIFIC ARTIFACTS FOR AUTOMATIC SOTA DISCOVERY

To describe the overall pipeline of ARTIFACTLINKER, we first formalize the problem definition of automatic discovery using the artifact-graph formulation. We then introduce our scalable solution, which addresses this problem via a two-stage *prediction–verification* framework.

## 4.1 DEFINITION OF AUTOMATIC SOTA DISCOVERY

Building on the artifact graph $\mathcal{G} = (\mathcal{V}, \mathcal{E})$ defined in the previous section, each observed edge $(m, d) \in \mathcal{E}$ is annotated with a performance score given by the metric function $\phi : \mathcal{V}_m \times \mathcal{V}_d \to \mathcal{K}$, with a realized score denoted as $\phi(m, d)$. The ultimate goal of automatic discovery is to identify a missing link $(m, d) \notin \mathcal{E}, m \in \mathcal{V}_m, d \in \mathcal{V}_d$ such that the predicted score under the same evaluation function $\phi(m, d)$ exceeds the best known performance on dataset $d$:

$$\phi(m, d) > \max_{(m', d) \in \mathcal{E}} \phi(m', d). \tag{1}$$

In other words, automatic SOTA discovery seeks model–benchmark pairs not yet connected in $\mathcal{G}$ but expected to advance the state of the art on $d$. The input of our proposed ARTIFACTLINKER framework is the artifact graph $\mathcal{G}$ while the target output is these state-of-the-art $(m, d)$ pairs.

---

[1]See details at https://huggingface.co/docs/hub/en/model-cards

## 4.2 SCALABLE LINK DISCOVERY FRAMEWORK

Because verification through explicit model evaluation is computationally expensive, scalability requires a mechanism to reduce the number of candidates before evaluation. To this end, we design a two-stage framework that factors the problem into two sub-problems: (1) *prediction* and (2) *verification*. In the prediction stage, we first produce a ranked set of candidate pairs $(m, d)$ by modeling a score function $s(m, d)$. From this ranking we derive the candidate set $\mathcal{C}$, which contains those pairs whose predicted scores are close to or surpass the current state of the art. In the verification stage, each candidate in $\mathcal{C}$ is executed and evaluated in practice, yielding the refined set $\mathcal{C}^*$ of validated state-of-the-art discoveries. This design focuses computational resources on the most promising candidates while ensuring correctness through real evaluation.

### 4.2.1 PREDICTION STAGE WITH SCORE FUNCTION MODELING

The goal of prediction is not only to estimate $\hat{\phi}(m, d)$ for missing edges but also to rank all candidate links by their discovery potential. Concretely, each prediction model defines a scoring function $s : \mathcal{V}_m \times \mathcal{V}_d \to \mathbb{R}$, which assigns each unseen pair $(m, d) \notin \mathcal{E}$ a predicted performance. For each model the objective is to approximate the score function $\hat{\phi}(m, d)$ and produce rankings consistent with observed edges. Since all of our prediction models utilize graph information, we formulate them using a general GNN-style notation. Formally, each score function is defined as:

$$\mathbf{h}_v^{(k+1)} = \text{AGG}^{(k)}\Big(\mathbf{h}_v^{(k)}, \{\mathbf{h}_u^{(k)}, e_{uv} : u \in \mathcal{N}(v)\}\Big), \quad s(m, d) = \psi\big(\mathbf{h}_m^{(L)}, \mathbf{h}_d^{(L)}\big), \quad (2)$$

where $\mathbf{h}_v^{(k)}$ is a model's internal representation of node $v$ after $k$ iterations aggregated over all representations $\mathbf{h}_u^{(k)}$ constructed from neighboring nodes $u \in \mathcal{N}(v)$. The final score is $s(m, d)$ is then defined over the final node representations $\mathbf{h}_m^{(L)}$ and $\mathbf{h}_d^{(L)}$ transformed via some function $\psi$. As we detail below, different instantiations of AGG and $\psi$ lead to different modeling strategies.

**LLM-based modeling**. These models are based on ordinary prompt-based frozen LLMs. Following the TextGNN formulation presented in Yu et al. (2024), we represent each $\mathbf{h}_v^{(k)}$ directly with text instead of embeddings. The aggregation function AGG is instantiated as a one-layer *concat* operator that serializes neighborhood information into a textual prompt:

$$\mathbf{h}_v^{(k+1)} = \text{CONCAT}\Big(\mathbf{h}_v^{(k)}, \{\mathbf{h}_u^{(k)}, e_{uv} : u \in \mathcal{N}(v)\}\Big), \quad s(m, d) = \text{LLM}\big(\mathbf{h}_m^{(L)}, \mathbf{h}_d^{(L)}\big). \quad (3)$$

Here, $\mathbf{h}_v^{(0)} = \mathbf{x}_v$ corresponds to the textual description of node $v$. The CONCAT function converts all neighbor attributes into natural language and concatenates them into a serialized context. The LLM then consumes these prompts for $m$ and $d$, jointly reasoning over their neighborhoods to predict $s(m, d)$. We typically adopt a single TextGNN layer due to the limited context window size of LLMs.

**GNN-based modeling**. Alternatively, we instantiate AGG using standard message passing in Graph Neural Networks. In this setting, node features are initialized as $\mathbf{h}_v^{(0)} = \text{EMBED}(\mathbf{x}_v)$ from the textual description $\mathbf{x}_v$, and representations are updated by:

$$\mathbf{h}_v^{(k+1)} = \sigma\Big(W^{(k)}\mathbf{h}_v^{(k)} + \text{AGG}^{(k)}\big(\{W^{(k)}\mathbf{h}_u^{(k)}, e_{uv} : u \in \mathcal{N}(v)\}\big)\Big), s(m, d) = \text{MLP}\Big([\mathbf{h}_m^{(L)} \| \mathbf{h}_d^{(L)}]\Big). \quad (4)$$

Here, $\text{AGG}^{(k)}(\cdot)$ can be implemented with classical GNN operators such as GATv2Conv (Brody et al., 2021), which applies dynamic attention weights to neighbor embeddings. The edge scoring function $s(m, d)$ is then computed by an MLP over the concatenation of the final node embeddings $\mathbf{h}_m^{(L)}$ and $\mathbf{h}_d^{(L)}$. In contrast to the LLM models above, this model is tuned using a training set of positive and negative links mined from the original artifact graph (see full details in Appendix. B.1)

**Ranking with score function**. Given different modeling choices of $s(m, d)$, the prediction stage first produces a ranked list of all unseen pairs $(m, d) \notin \mathcal{E}$, ordered by their predicted score $s(m, d)$. This ranking reflects the discovery potential of each candidate, with higher scores indicating stronger likelihood of advancing the state of the art.

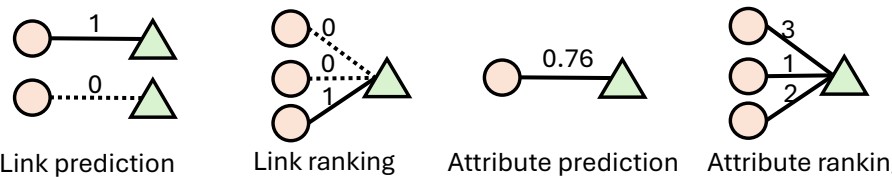

Figure 2: **Four types of evaluation tasks for prediction**. From left to right: (1) 0/1 binary link prediction between existing edges and negative sampled edges; (2) link ranking between existing edges and negative sampled edges; (3) attribute prediction to conduct regression on a single existing edge; (4) attribute ranking between edges with smaller attributes and edges with higher attributes.

From this ranked list, we further define the set of promising candidates as

$$\mathcal{C} = \left\{ (m,d) \,\middle|\, (m,d) \notin \mathcal{E},\ m \in \mathcal{V}_m,\ d \in \mathcal{V}_d,\ s(m,d) \ \geq\ \max_{(m',d) \in \mathcal{E}} \phi(m',d)\ -\ \delta \right\}, \quad (5)$$

where $\delta \geq 0$ is a tolerance parameter and $\phi(m',d)$ is the observed evaluation score of model $m'$ on dataset $d$. Larger $\delta$ indicates higher verification cost but the greater possibility for finding the state-of-the-art performance. In other words, the score function $s(m,d)$ serves a dual purpose: it provides a global ranking over all candidates, and it defines a thresholded subset $\mathcal{C}$ to guide the verification stage by focusing computation on the most promising discoveries.

### 4.2.2 VERIFICATION STAGE WITH MULTI-STAGE PLANNING

Given the candidate set $\mathcal{C}$ with a relatively small size, the goal of verification is to execute and collect the concrete evaluation results. We verify each pair $(m,d)$ with an agentic workflow, RE-ACTLINKER, that plans, generates, executes, and refines code for automatic benchmarking at scale. REACTLINKER performs *node-first* validation before any edge-level run via three sub-components: (1) *benchmark checker* $\mathrm{check}_d$ (resolve dataset on HuggingFace, validate version/splits, load a shard, and materialize examples), (2) *model checker* $\mathrm{check}_m$ (resolve model, verify dependencies/device, and dry-forward to confirm I/O contracts), and (3) *metric checker* $\mathrm{check}_k$ (instantiate the evaluation metric and sanity-check on synthetic $(\hat{y}, y)$). Each stage within REACTLINKER is a ReAct loop (Yao et al., 2023). Only if all three checks pass does REACTLINKER compose the pipeline (model $\rightarrow$ dataset $\rightarrow$ metric) and run the edge evaluation to obtain a real score:

$$s^*(m,d) \ =\ \text{REACTLINKER}\big(\mathbf{x}_m, \mathbf{x}_d\big).$$

**SOTA selection with execution.** We keep only candidates that pass all node checks, execute successfully, and beat the best known score on $d$:

$$\mathcal{C}^* \ =\ \left\{ (m,d) \,\middle|\, (m,d) \in \mathcal{C}\ (\text{Eq. 5}),\ s^*(m,d) \geq \max_{(m',d) \in \mathcal{E}} \phi(m',d) \right\}.$$

This multi-stage agnetic workflow design isolates configuration errors early, enables caching of validated nodes across pairs, and avoids wasted full runs—making verification robust and scalable under the standardized structure of HuggingFace artifacts.

## 5 EVALUATING LINK DISCOVERY WITH EDGE MASKING

In Section §4, we described how ARTIFACTLINKER is designed and used to perform automatic SOTA link discovery. However, beyond training and inference, we also need systematic evaluation tasks for both the prediction and verification stages. To this end, we design a mask-and-predict/verify protocol based on the artifact graph. Instead of running inference in the real world, we directly sample missing links $(m,d) \notin \mathcal{E}$ and evaluate them on ARTIFACTBENCH.

For evaluation, we consider two forms of masking. (i) *Edge masking*: a subset of observed edges $\mathcal{E}_{\mathrm{mask}} \subset \mathcal{E}$ is hidden from the graph, requiring the model to recover their existence. (ii) *Attribute masking*: for another subset of edges, we keep the edge structure visible but hide their associated attributes (e.g., evaluation scores), requiring the model to predict or rank the missing values.

**Prediction evaluation**. Given the masked graph $\mathcal{G} \setminus \mathcal{E}_{\text{mask}}$, we benchmark prediction performance with four tasks, as shown in Figure 2. (1) *Link prediction (edge masking)*: determine whether a masked edge $(m, d) \in \mathcal{E}_{\text{mask}}$ exists, i.e., $\hat{y}_{m,d} = \mathbf{1}[s(m, d) > \tau]$, where $s(m, d)$ is the predicted score function. (2) *Link ranking (edge masking)*: for each dataset $d$, rank all candidate models $m$ by $s(m, d)$ and evaluate the rank of the true masked edges $(m, d) \in \mathcal{E}_{\text{mask}}$. (3) *Attribute prediction (attribute masking)*: for each edge $(m, d)$ with hidden attributes, predict its score $\hat{\phi}(m, d)$. (4) *Attribute ranking (attribute masking)*: for each dataset $d$, rank all candidate models $m$ by predicted attributes $\hat{\phi}(m, d)$ and compare this ordering against ground-truth scores.

**Verification evaluation**. Finally, we define the *reproduction task* in the verification stage, which also corresponds to attribute masking. Here, the coding agent re-evaluates edges $(m, d)$ with hidden attributes to measure consistency between predicted scores $s(m, d)$ and execution-based results $\phi^*(m, d)$. Formally, consistency is satisfied if $\big| s(m, d) - \phi^*(m, d) \big| \leq \delta$, where $\delta$ is a tolerance parameter defined for the same metric. This criterion checks whether prediction and execution agree up to an acceptable margin, ensuring that verification faithfully reproduces benchmarked performance.

# 6 EXPERIMENTAL SETTINGS

**Prediction task settings**. We split the overall artifact graph into train, development, and test edge splits based on a 70%-10%-20% ratio. This leads to 1,448 edges in the train split, 206 edges in the development split, and 413 edges in the test split.

**Prediction baseline settings**. Besides the LLM-based TextGNN settings and GNN-based settings, we include multiple baseline settings for comparison: (1) *random*, where for both the prediction and ranking tasks we generate random results as a baseline; (2) *metadata*, where we directly use metadata of the artifacts such as downloading times as a feature for prediction; (3) *node degree*, where we utilize graph-related structural features like node degree for prediction; (4) *LLM baselines*, where we only provide the model information and benchmark information to an LLM (we use GPT-4o (OpenAI et al., 2024), Qwen2.5-72B (Qwen et al., 2025), and o3) and ask them to directly predict.

**Verification task settings**. We construct the verification benchmark by selecting model–benchmark pairs from the artifact graph that include reported scores (e.g., F1, accuracy, BLEU, ROUGE). For each pair, we test whether the reported result can be reproduced through execution-based evaluation, given the model and benchmark specifications. To capture varying levels of difficulty, we define two splits: (i) a *hard split*, consisting of large models, complex benchmarks, and metrics that are more challenging to reproduce (e.g., MMLU with Mistral-7B); and (ii) an *easy split*, covering straightforward settings such as evaluating BERT-based models on SST-2 with accuracy.

**Verification baseline settings**. As baselines, we adopt ReAct (Yao et al., 2023), which employs a single coding agent that iteratively debugs execution errors. Importantly, this baseline does not include HuggingFace-specific design choices, but simply uses model and benchmark descriptions as inputs. For a fair comparison, both ReAct and our proposed ReActLinker use GPT-4o as the underlying backbone model.

# 7 EXPERIMENTAL RESULTS

## 7.1 EXPERIMENTAL RESULTS ON PREDICTION TASKS

**Structural information from the graph is useful for prediction**. Across all tasks—including link prediction (Table 1), link ranking (Table 2), attribute prediction (Table 3), and attribute ranking (Table 4)—we find that incorporating neighborhood information into the LLM-based score function substantially improves performance. The gains are especially pronounced for link prediction and attribute prediction: for example, adding 1-hop neighborhood information increases GPT-4o's F1 from 64.6 to 83.4 in link prediction and reduces MAE from 8.9 to 5.0 in attribute prediction. Improvements on ranking tasks are smaller, likely because LLMs already receive rich contextual information about candidate items. Nevertheless, the effect is consistent across different backbone models (GPT-4o, Qwen2.5-72B, and o3), validating the utility of the artifact graph.

**GNN-based models achieve strong performance**. Beyond LLM-based modeling, we also evaluate GNN-based methods using a simple 3-layer GATv2Conv. Node embeddings are initialized with

Table 1: **Link prediction results.** We randomly sample an equal number of negative samples from the artifact graph for binary 0/1 link prediction and keep existing edges as positive samples. More information about baselines is available in Section §6.

| Method | F1 | P | R | Acc |
|---|---|---|---|---|
| Random | 50.0 | 50.0 | 50.0 | 50.0 |
| Metadata | 52.2 | 58.6 | 47.0 | 56.9 |
| Node degree | 87.2 | 91.5 | 83.3 | 87.8 |
| GPT-4o | 64.6 | 95.2 | 48.9 | 73.2 |
| Qwen2.5-72B | 77.3 | 94.6 | 65.4 | 80.8 |
| o3 | 58.8 | 96.7 | 42.3 | 70.4 |
| GPT-4o+graph | 83.4 | 90.3 | 77.5 | 84.6 |
| Qwen2.5+graph | 77.3 | 94.6 | 65.4 | 80.8 |
| o3+graph | 63.1 | 97.7 | 46.6 | 72.8 |
| GATv2Conv | **88.4** | 95.8 | 82.1 | **89.2** |

Table 2: **Link ranking results.** We build a ranking task for each benchmark and sample 10 more as negative candidates for each benchmark. More information about baselines is available in Section §6. R@$k$ refers to Recall@$k$ and P@$k$ refers to Precision@$k$.

| Method | R@1 | R@5 | P@1 | P@5 |
|---|---|---|---|---|
| Random | 6.9 | 38.9 | 20.0 | 21.2 |
| Metadata | 6.7 | 38.1 | 17.8 | 19.4 |
| Node degree | 17.7 | 61.2 | 41.1 | 33.3 |
| GPT-4o | 49.2 | 86.0 | 82.8 | 45.0 |
| Qwen2.5-72B | 47.4 | 86.3 | 83.2 | 44.8 |
| o3 | 52.2 | 87.2 | 87.6 | 45.6 |
| GPT-4o+graph | 53.7 | 90.6 | 89.9 | 47.4 |
| Qwen2.5+graph | 54.8 | **90.7** | 88.9 | 44.1 |
| o3+graph | **54.1** | 89.0 | **91.7** | **47.6** |
| GATv2Conv | 36.9 | 84.2 | 55.8 | 34.0 |

Voyage-3[2] embeddings. Surprisingly, this lightweight GNN achieves competitive or even state-of-the-art performance. For instance, it matches GPT-4o on attribute prediction MAPE (8.9 vs. 9.0) and outperforms all methods on link prediction. While GNNs are relatively weaker on ranking tasks compared with LLM-based approaches, they remain far stronger than baseline heuristics. These results suggest that structural patterns in the research community graph encode valuable signals for predicting model–dataset performance.

**Overall: prediction and ranking tasks defined on the artifact graph are valuable but challenging**. Our experiments show that, apart from link prediction (where random negative sampling is used), other sub-tasks—link ranking, attribute prediction, and attribute ranking—still leave substantial room for improvement. We argue that progress on this benchmark will be crucial for enabling scalable automatic discovery.

### 7.2 EXPERIMENTAL RESULTS ON VERIFICATION TASKS

**Specialized workflows improve robustness under hard cases**. As shown in Figure 3, specialized multi-stage workflows influence both the execution rate and successful reproduction rate. While execution rate may decrease a lot under the easy cases (from 81.0 to 63.0)—since the agent explores more possibilities and can become stuck at certain stages—the overall reproduction success improves in harder scenarios. In particular, our proposed ReActLinker can handle stage-specific errors independently, leading to substantially better performance in challenging cases.

**Overall: verification on the artifact graph is valuable but cannot be solved trivially**. Despite HuggingFace providing a convenient infrastructure for running models and benchmarks, current coding agents still struggle to reliably reproduce reported results. Our findings highlight both the promise and the limitations of verification tasks on the artifact graph: they are valuable for evaluating agent capabilities, yet they remain unsolved in practice. Building more robust auto-evaluation agents for HuggingFace thus represents an important future direction for the ML research community.

## 8 DISCUSSION

**Q1: What is the role of node attributes in the artifact graph?** Node attributes provide essential context about models and benchmarks beyond evaluation results. We extract attributes such as training methods, architectures, and benchmark statistics using LLMs, which serve two purposes. First, for LLM-based predictors, these attributes activate relevant parameter knowledge and support reasoning by analogy (e.g., if two models share architecture and training setup, their expected performance is similar). For instance, an LLM may reason: "*Since model A achieved 0.979 accuracy on CIFAR-10, model B with the same architecture and training regime is expected to reach 0.978.*"

---

[2]https://docs.voyageai.com/docs/embeddings

Table 3: **Attribute prediction results**. We focus on predicting the attributes on existing edges. MSE refers to Mean Squared Error, MAE refers to Mean Absolute Error, MAPE refers to Mean Absolute Percentage Error, and $R^2$ refers to Coefficient of Determination.

| Method | MSE | MAE | MAPE | $R^2$ |
|---|---|---|---|---|
| Global mean | 3.48 | 14.9 | 26.3 | -0.5 |
| Local mean | 1.94 | 8.6 | 14.7 | 43.9 |
| GPT-4o | 1.84 | 8.9 | 16.6 | 46.9 |
| Qwen2.5-72B | 2.37 | 10.5 | 20.7 | 31.5 |
| o3 | 2.47 | 9.1 | 13.9 | 28.7 |
| GPT-4o+graph | 0.90 | 5.0 | 9.0 | **74.1** |
| Qwen2.5+graph | 1.23 | 6.5 | 11.9 | 70.2 |
| o3+graph | 1.05 | **4.1** | **6.8** | 69.3 |
| GATv2Conv | 1.32 | 6.0 | 8.9 | 49.3 |

Table 4: **Attribute ranking results**. Similar to link ranking tasks, we build an attribute ranking task for each benchmark. Only models reporting the same metric are included for ranking. We focus on ranking the attributes on existing edges instead of sampling negative ones.

| Method | NDCG@1 | Hit@1 | Hit@5 |
|---|---|---|---|
| Random | 27.4 | 58.5 | 47.8 |
| Node degree | 36.7 | 11.3 | 56.5 |
| GPT-4o | 61.8 | **43.4** | 82.6 |
| Qwen2.5-72B | 58.6 | 35.9 | 82.6 |
| o3 | 58.1 | 36.5 | 77.3 |
| GPT-4o+graph | 62.0 | 35.9 | **87.0** |
| Qwen2.5+graph | **62.3** | 38.5 | 81.8 |
| o3+graph | 60.5 | 37.7 | 78.3 |
| GATv2Conv | 41.8 | 31.0 | 40.0 |

Second, for GNN-based predictors, textual attributes are embedded using state-of-the-art embedding models (e.g., voyage-3) to capture similarity between models and datasets. Empirically, when node features are initialized randomly instead of using text embeddings, the MSE of attribute prediction worsens from 0.01 to 0.22—showing that attributes are crucial for learning meaningful patterns.

**Q2: Why are GNNs useful for predictions?** GNNs are effective because machine learning communities exhibit strong and predictable structural patterns. Popular benchmarks (e.g., CoNLL2003 for NER or CNN/DailyMail for summarization) attract consistent evaluation across related models, while models sharing similar architectures or training methods tend to achieve similar performance. The extreme central point for ImageNet-1k in Figure 4 is a strong evidence for that. These structural regularities allow GNNs to achieve performance close to LLM-based predictors despite relying only on graph structure and local attributes and without complicated reasoning. In short, GNNs succeed because the artifact graph encodes community-wide regularities that are highly predictive of links.

## 9 CASE STUDY

To illustrate the value of ARTIFACTLINKER, we run it on SQuAD-v2[3]. Because unseen-edge discovery is difficult to benchmark in a controlled setting, we provide a realistic case study on SQuAD-v2, showing how ARTIFACTLINKER retrieves, ranks, and verifies on unseen but promising model–benchmark pairs. We first sample candidate models for link prediction and then perform attribute prediction on the exact-match (EM) metric. Our predictor ranks *Llama-3-70B-Instruct* highest with a predicted EM of 0.86, consistent with reported results. Moreover, executing smaller high-ranked candidates reveals that *SGPT-125M* (tuned on MS MARCO) achieves an EM of 0.41 on SQuAD-v2—surprisingly strong for a 125M model not explicitly trained on SQuAD-v2. These verified findings provide actionable insight even when they are not SOTA. **However, large-scale verification remains the primary bottleneck:** end-to-end execution is constrained by environment setup and dependency drift, heterogeneous evaluation harnesses, API/rate limits, flaky runs, and substantial compute and orchestration overheads. In practice, fully automated, high-throughput verification at community scale is still an open systems challenge.

## 10 CONCLUSION

In this paper, we introduce ARTIFACTLINKER, a framework for turning HuggingFace into an engine for automatic SOTA discovery. By modeling models and benchmarks as an artifact graph, we define the task of discovering valuable missing links that reveal promising model–benchmark interactions. To address this, we propose a two-stage *predict-and-verify* framework, where prediction identifies

---

[3]`https://huggingface.co/datasets/rajpurkar/squad_v2`

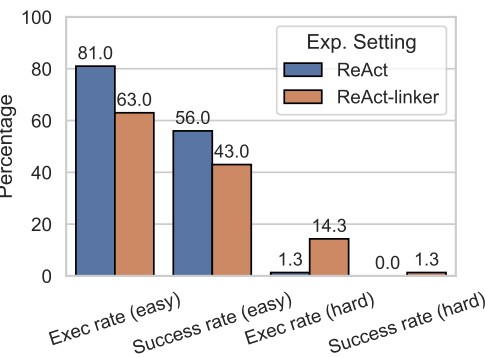 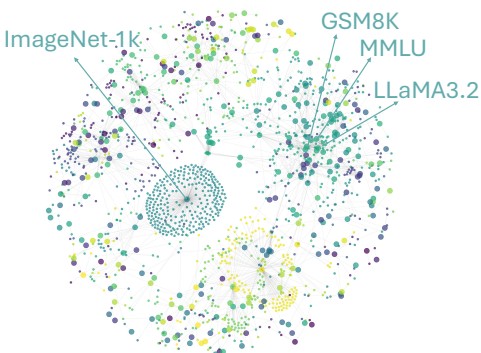

Figure 3: **Success rate and execution rate for verification tasks on the artifact graph**. Execution rate indicates whether code snippets generated by agents can be executed or not. Success rate indicates whether code snippets generated by agents can provide close performance compared with reported ones.

Figure 4: **Visualization of the artifact graph**. Larger dots represent the benchmarks and the smaller dots represent the models. Both of them are collected from the HuggingFace Hub. Different node colors represent models and datasets in different domains. A famous benchmark like ImageNet-1k is evaluated by multiple models.

promising candidates and verification conducts execution-based experiments to validate their performance. We further develop ARTIFACTBENCH, which provides systematic tasks for assessing both prediction and verification. Our results show that graph-based prediction is effective for surfacing high-potential links, and that automated verification can reproduce many benchmark results without human intervention. Looking ahead, we envision extending this direction to dynamically grow the artifact graph by incorporating new models and benchmarks, enabling continual and scalable scientific discovery.

## REPRODUCIBILITY STATEMENT

All implementation details of our method and baselines are given in Section §6 and Appendix §B. We will release our benchmark and the codebase at the time of publication.

## ETHICS STATEMENT

Our work focuses on building an automatic discovery framework over open-source artifacts available on HuggingFace. All experiments are conducted exclusively on publicly released models and benchmarks that are freely accessible to the research community. We do not introduce any new human or sensitive data, nor do we attempt to deanonymize or misuse existing artifacts.

The goal of our framework is to advance automated, reproducible, and scalable scientific discovery, and to help researchers more efficiently identify promising model–benchmark interactions. Nevertheless, we acknowledge that automated benchmarking may propagate existing biases and limitations present in the underlying models and datasets. To mitigate this, we emphasize transparency in data collection and reproducibility in our verification pipeline. We encourage the community to view our work as a step toward building more reliable, equitable, and responsible auto-discovery systems, rather than a replacement for human oversight.

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

## A  THE USE OF LARGE LANGUAGE MODELS (LLMS)

We used ChatGPT as a writing assistant to help us write part of the paper. Additionally, we utilize the power of CodePilot to help us code faster. However, all the AI-generated writing and coding components are manually checked and modified. There is no full AI-generated content in the paper.

## B  EXPERIMENTAL DETAILS

### B.1  GNN-BASED MODELING AND TRAINING DETAILS

In this section, we describe the architecture of our GNN-based model and provide details on its training setup for both link prediction and attribute regression tasks.

### B.2  MODEL ARCHITECTURE

Our link prediction model, `GNNLinkPredictor`, combines a graph neural network encoder with a flexible edge-scoring decoder. The encoder is based on GATv2 and stacks multiple convolutional layers with `GraphNorm`, `PReLU` activations, and feature dropout. Residual connections are added when input and output dimensions match, and self-loops as well as edge dropout can be applied for stability. To integrate information across layers, we adopt Jumping Knowledge (JK), supporting concatenation with projection, element-wise max pooling, or using the last layer only. The resulting node embeddings are passed into the edge decoder (`EdgePredictor`), which supports various scoring functions, including dot product, cosine similarity, bilinear transformation, linear projection on concatenated features, and a two-layer MLP. This design balances expressiveness and flexibility: the encoder captures rich multi-hop structure, while the decoder adapts to diverse relational patterns.

### B.3  TRAINING FOR LINK PREDICTION

For binary edge classification, we train our GNN model over positive and negative edges sampled from the artifact graph, using pre-computed node embeddings as input. We optimize with AdamW (learning rate $5 \times 10^{-3}$, weight decay $10^{-4}$) and use a ReduceLROnPlateau scheduler (patience 10, decay factor 0.8, minimum learning rate $10^{-6}$). Training runs for up to 300 epochs with early stopping (patience 40) based on validation AUC, and class imbalance is corrected by weighting the binary cross-entropy loss with $\frac{N_{\text{neg}}}{N_{\text{pos}}}$. Mixed-precision training is enabled on GPU. We evaluate every 50 epochs, select the best checkpoint by validation AUC, and report accuracy, precision, recall, F1, ROC-AUC, and average precision on the test set. Unless otherwise noted, we use 64 hidden dimensions, 3 layers, 3 heads, dropout 0.2, and seed 42.

### B.4  TRAINING FOR ATTRIBUTE REGRESSION

For continuous edge-attribute prediction, we train on positive edges with metric values, normalizing values greater than 1 to $[0, 1]$. Node embeddings are pre-computed and message passing uses the training graph's edges. The model regresses edge scores directly in logit space: ground-truth targets are transformed via $\text{logit}(y) = \log \frac{y}{1-y}$, and MSE is computed against the raw decoder logits. This stabilizes training near the boundaries. We use Adam (learning rate 0.005, weight decay $10^{-5}$), Xavier-uniform initialization for linear layers, dropout 0.2, hidden size 128, 3 layers, and 8 heads (GATv2 backbone). Training runs up to 500 epochs, with frequent evaluation early (every 10 epochs for the first 50, then every 25), and the best checkpoint is selected by lowest validation MSE. At evaluation, logits are clipped to $[-10, 10]$ and mapped through sigmoid to report MSE, MAE, RMSE, $R^2$, and MAPE.

### B.5  LLM-BASED MODELING DETAILS

We directly rely on in-context learning for LLM-based modeling without training. We provide examples of our prompts for different tasks below:

Listing 1: LLM with graph link prediction example

Given a machine learning model named 'tensorblock/bloomz-3b-GGUF' and a
    dataset named 'SetFit/rte'.

More information about this model: The model 'bloomz-3b1' is a
    multilingual text-generation model based on the BLOOM architecture.
    It is fine-tuned on a variety of datasets such as the BigScience xP3,
     which contributes to its ability to handle multiple languages and
    tasks. The model supports a wide range of languages including English
    , Spanish, French, Chinese, and many others, making it highly
    versatile for global applications. It is also capable of
    understanding and generating programming code in several languages,
    like C++, Java, Python, and more. This flexibility is reflected in
    its usage across diverse tasks, from sentiment analysis and question
    answering to natural language inference and program synthesis. The
    model operates under the bigscience-bloom-rail-1.0 license, ensuring
    its use in accordance with open research and collaboration principles
    .

More information about this dataset: The Glue RTE dataset is derived from
     the Recognizing Textual Entailment (RTE) task, a subset of the GLUE
    benchmark. It is designed to evaluate models on the task of natural
    language inference, which involves determining if one sentence
    logically follows from another. In the ported version hosted on
    Hugging Face's platform, it retains its original purpose but
    introduces some modifications for easier processing. Notably, the
    columns originally named 'sentence1' and 'sentence2' have been
    renamed to 'text1' and 'text2' respectively. This dataset is often
    used to train and evaluate machine learning models on their ability
    to understand and process natural language, particularly in contexts
    where discerning relationships between two text statements is crucial
    . One important characteristic of the dataset is that its test split
    is unlabeled, with all entries in the label column set to -1. This is
     in line with many benchmark datasets where the test labels are
    withheld to prevent overfitting and encourage robust model evaluation
    .

There are other models that are evaluated on the dataset to judge whether
     the model and dataset are connected:
- 42dot/42dot_LLM-PLM-1.3B: 42dot LLM-PLM is a pre-trained language model
     developed by 42dot, designed to handle both Korean and English text.
     It is part of the 42dot LLM series, leveraging a 1.3 billion
    parameter configuration based on a Transformer decoder architecture,
    akin to LLaMA 2. The model is trained with a corpus of diverse text
    sources, both in Korean and English, aimed at providing a versatile
    foundation for various natural language tasks. The architecture
    consists of 24 layers, 32 attention heads, a hidden size of 2048, and
     an FFN size of 5632. Pre-training involved 49,000 GPU hours using
    NVIDIA A100, with specific hyperparameters such as a global batch
    size of 4 million tokens and a learning rate of 4E-4. A Byte-level
    BPE tokenizer was developed from scratch to support the model,
    featuring a vocabulary size of approximately 50,000 tokens.
- CobraMamba/mamba-gpt-3b: The Mamba GPT-3B model is a fine-tuned version
     of the open-lama model, providing enhanced performance across
    several evaluation subtasks. It stands out as the best-performing 3B
    model, achieving performance metrics comparable to the larger llama-7
    b model. This model utilizes the transformers library and has been
    optimized for GPU usage, requiring the installation of specific
    versions of the transformers, accelerate, and torch libraries. The
    Mamba GPT-3B is designed to generate text with a focus on maintaining
     coherence and context, making it suitable for various applications
    that require advanced language processing capabilities.
- ModelCloud/Meta-Llama-3.1-8B-Instruct-gptq-4bit: This model is a
    quantized version of Llama-3.1 with 8 billion parameters, created
    using the GPTQModel framework. It utilizes a 4-bit precision format,

which allows for efficient storage and computation without significant loss of model performance. The model's configuration includes a group size of 128, symmetric quantization, and true sequential processing. These settings are optimized to maintain a balance between model accuracy and computational efficiency. The quantization process reduces the model's footprint, making it suitable for deployment in environments with limited resources, while still providing robust performance on various tasks. The use of the GPTQ quantization method ensures that the model can handle different types of data inputs efficiently.

- ibm-research/ColD-Fusion: The ColD Fusion model is a finetuned language model that builds upon the RoBERTa base architecture. It was trained on 35 diverse datasets and is designed to serve as a robust base model for various natural language processing tasks. The model leverages a novel approach called ColD Fusion, which allows for multitask learning benefits through distributed computation. Unlike traditional multitask learning that requires simultaneous access to all datasets and substantial computational resources, ColD Fusion operates with limited communication and no data sharing. This enables the recycling of finetuned models to iteratively enhance the pretrained model. The resulting model achieves strong performance across the datasets it was trained on and serves as an excellent starting point for finetuning on new datasets. The model demonstrates a 2.45-point improvement over RoBERTa base without architectural changes and outperforms previous multitask models. For detailed insights, refer to the paper linked in the README.

- PygmalionAI/metharme-1.3b: Metharme 1.3B is an instruction-tuned language model based on EleutherAI's Pythia 1.4B Deduped. It is designed specifically for conversation, roleplaying, and storywriting with a focus on fictional contexts. This model can be guided using natural language like other instruct models. It has been fine-tuned using a mixture of regular instruction data, alongside roleplay and fictional stories, with synthetically generated instructions. Metharme 1.3B utilizes a unique prompting system with three role tokens: `<|system|>`, `<|user|>`, and `<|model|>`. The `<|system|>` token can provide background information, `<|user|>` indicates user input, and `<|model|>` signifies the model's generated response. These tokens can be chained to create conversation history. The model is suitable for entertainment purposes, particularly in fictional writing, and it is not optimized for safety or harmlessness. It may produce offensive or socially unacceptable content and is not reliable for factual accuracy.

- tensorblock/bloomz-7b1-mt-GGUF: The model 'bloomz-7b1-mt' is a multilingual text-generation model based on the BigScience Bloom architecture. It supports a wide array of languages including common ones like English, Spanish, and Chinese, as well as less common languages such as Fon, Twi, and Tsonga. Additionally, it is capable of understanding and generating code in several programming languages, including Python, JavaScript, and C++. The model is designed to handle various natural language processing tasks such as coreference resolution, natural language inference, and sentence completion. It uses datasets like Winogrande, XWinograd, and XNLI, among others, to evaluate its performance across different tasks. The model is distributed under the bigscience-bloom-rail-1.0 license and is part of the 'bigscience/xP3mt' dataset, reflecting its broad multilingual and multitask focus. Its applications can range from sentiment analysis and query suggestion to the generation of creative content such as fairy tales and fables.

Please predict whether this model should be evaluated on this dataset. Provide your answer as a JSON object with two keys: 'prediction' (a boolean, true or false) and 'reason' (a brief explanation of your reasoning).

Listing 2: LLM with graph attribute prediction example

```
Given a machine learning model named 'tasksource/ModernBERT-large-nli'
    and a dataset named 'stanfordnlp/snli'.

More information about this model: ModernBERT is a multi-task fine-tuned
    model specifically designed for natural language inference (NLI)
    tasks. It has been trained on a diverse set of NLI datasets including
     MNLI, ANLI, SICK, WANLI, doc-nli, LingNLI, FOLIO, FOL-NLI, LogicNLI,
     and Label-NLI among others. The training was conducted over 200,000
    steps using an Nvidia A30 GPU, resulting in a powerful model for
    reasoning tasks. ModernBERT surpasses the performance of models like
    llama 3.1 8B Instruct in specific tasks such as ANLI and FOLIO. It
    excels in long context reasoning, sentiment analysis, and zero-shot
    classification with new labels. The model is versatile, offering
    significant potential for fine-tuning on single tasks like SST, but
    it is particularly effective out-of-the-box for zero-shot
    classification and NLI tasks.

More information about this dataset: The Stanford Natural Language Infer(
    Pdb) ence (SNLI) corpus is a comprehensive dataset designed to
    support the task of natural language inference (NLI), also known as
    recognizing textual entailment (RTE). Version 1.0 of the dataset
    comprises 570,000 human-written sentence pairs labeled with
    entailment, contradiction, and neutral. These labels are crucial for
    training and evaluating models that aim to determine the inference
    relationship between two short texts. The SNLI dataset is an
    essential resource for developing systems that can understand and
    predict semantic relationships in natural language, making it a
    benchmark for text representation learning methodologies. The dataset
     was created through crowdsourcing efforts, with premises originating
     from the Flickr 30k and VisualGenome corpora, and hypotheses
    generated by crowdworkers on Amazon Mechanical Turk. Each sentence
    pair includes a premise and a hypothesis, and the task is to classify
     the relationship between them. The dataset provides a balanced
    classification challenge, offering a rich source of data for training
     machine learning models in natural language processing applications.

tasksource/ModernBERT-large-nli's performance on other datasets:
- pietrolesci/nli_fever: accuracy: 0.780 (info: The dataset in question
    is a modification of the FEVER dataset, tailored for Natural Language
     Inference (NLI) research. Originally, the FEVER dataset consisted of
     claims derived from Wikipedia, each paired with a label indicating
    the veracity of the claim. However, this setup did not align with
    standard NLI tasks which typically involve a pair of sequences (e.g.,
     premise and hypothesis) mapped to a label. To bridge this gap and
    facilitate NLI research, the creators of this dataset have
    reformatted it to pair claims with textual evidence, thus converting
    it into a pair-of-sequences to label dataset. This transformation
    enables the application of NLI models on the FEVER dataset. The
    labels are mapped using predefined categories, where 'SUPPORTS' is
    mapped to entailment, 'NOT ENOUGH INFO' to neutral, and 'REFUTES' to
    contradiction. Additionally, the dataset includes a 'verifiable'
    column encoded to indicate whether a claim can be verified. Despite
    these modifications, the dataset maintains consistency with the
    original FEVER dataset, ensuring reliability and validity for
    research purposes.)
- yale-nlp/FOLIO: accuracy: 0.710 (info: The dataset is designed to
    provide a comprehensive collection of data relevant to the training
    and evaluation of machine learning models across various applications
    . It comprises a wide range of data types and structures, ensuring
    versatility and scalability for different research and development
```

purposes. The dataset is curated to support both supervised and
unsupervised learning tasks, facilitating experimentation with
classification, regression, clustering, and more advanced machine
learning methodologies. Each data entry in the dataset is
meticulously labeled and documented, providing clear context and
metadata to aid in the effective utilization of the data. The dataset
is continually updated to incorporate the latest advancements and
feedback from the user community, ensuring it remains relevant and
useful. It is distributed under the MIT license, allowing for broad
usage, adaptation, and distribution, making it an ideal resource for
both academic and commercial projects. The dataset's accessibility
and comprehensiveness make it a valuable asset for data scientists,
researchers, and developers aiming to advance machine learning
capabilities.)
- rediska0123/puzzte: accuracy: 0.590 (info: The dataset is designed to
serve as a resource for binary question-answering models. It consists
of questions that require a true or false answer. The main features
of the dataset include a 'question' field, which contains the text of
the question itself, and an 'answer' field, which is a boolean
indicating whether the answer is true or false. The dataset is
available in a single configuration named 'default' and contains a
single split, 'train', which includes 911 examples. The total dataset
size is 242,839 bytes, while the download size is 64,267 bytes.)
- causal-nlp/CLadder: accuracy: 0.890 (info: The dataset consists of
multiple configuration files specifying different data splits. In the
provided configurations, there are two main data files: 'full_v1.5
_default' and 'full_v1'. These files are likely CSV formatted and are
stored in the 'data' directory. The 'default' configuration suggests
that it might be the primary or recommended setup for working with
the dataset. Each configuration is defined under the 'configs' key,
with each containing a 'config_name' and associated 'data_files'. The
data files are specified with a 'split' name and the corresponding
file path. This setup allows users to select different versions or
subsets of the data depending on their use case, facilitating
experiments with varied data distributions or updated datasets. Such
configuration flexibility is crucial for testing machine learning
models under different conditions or for ensuring reproducibility in
research.)
- MoritzLaurer/dataset_train_nli: accuracy: 0.950 (info: The dataset
comprises multiple features including 'text', 'hypothesis', 'labels',
'task_name', and 'label_text'. The 'labels' feature is a class label
with two possible values: 'entailment' (0) and 'not_entailment' (1).
This dataset is primarily used for natural language inference tasks,
where the goal is to determine if a given hypothesis is entailed by
the premise text. The data is organized into a single configuration
named 'default' and focuses on a training split. The training data
consists of 1,018,733 examples, occupying 315,017,473 bytes. The
total download size for the dataset is approximately 206 MB. This
structured setup allows for easy implementation of machine learning
models aimed at understanding the entailment relationship between
pairs of sentences.)
- tasksource/ruletaker: accuracy: 0.990 (info: The 'ruletaker' dataset is
designed to evaluate the ability of transformer models to perform
reasoning over natural language. This dataset consists of synthetic
data generated to simulate logical reasoning tasks using controlled
language. The primary features of the dataset include 'context', '
question', 'label', and 'config', each represented as a string, which
are structured to test the model's capacity to infer and deduce
truths from given premises. The dataset is provided in three splits:
'train' with 480,152 examples, 'dev' with 75,872 examples, and 'test'
with 151,911 examples, collectively requiring a dataset size of
372,450,135 bytes. It is licensed under the Apache-2.0 license and
primarily intended for English language processing. The dataset is
based on the work presented in the paper 'Transformers as Soft
Reasoners over Language' by Peter Clark, Oyvind Tafjord, and Kyle

```
        Richardson, which was presented at the Twenty-Ninth International
        Joint Conference on Artificial Intelligence in 2020.)
- tasksource/babi_nli: accuracy: 0.980 (info: The babi_nli dataset is
    designed for natural language inference tasks, specifically focusing
    on logical reasoning capabilities. This dataset supports various
    logic-based tasks by providing pairs of premises and hypotheses, each
     labeled as either 'entailed' or 'not-entailed.' The dataset is
    monolingual, using English language text, and is created through a
    combination of expert-generated annotations and crowdsourcing efforts
    . It falls under the size category with examples ranging between
    1,000 and 10,000. The dataset offers multiple configurations, each
    targeting different aspects of logical reasoning such as agents'
    motivations, coreference, deduction, induction, and several others.
    Each configuration provides its own train, validation, and test
    splits, with varying dataset sizes and download sizes. The dataset
    can be used to evaluate models on their ability to perform natural
    language inference tasks, an essential component of many modern
    applications in natural language processing.)
stanfordnlp/snli's performance with other models:
- tasksource/ModernBERT-base-nli: accuracy: 0.830 (info: ModernBERT is a
    multi-task fine-tuned transformer model specifically designed for
    natural language inference (NLI) tasks. It incorporates datasets such
     as MNLI, ANLI, SICK, WANLI, doc-nli, LingNLI, FOLIO, FOL-NLI,
    LogicNLI, and Label-NLI, among others. The model is an 'instruct'
    version, optimized for tasks requiring reasoning and zero-shot
    classification. ModernBERT's training regime involved 200,000 steps
    on an Nvidia A30 GPU, enhancing its capabilities in long-context
    reasoning and sentiment analysis. It surpasses other models like
    llama 3.1 8B Instruct in specific NLI tasks such as ANLI and FOLIO.
    Though additional performance improvements can be achieved through
    single-task fine-tuning (e.g., SST), the current version excels in
    zero-shot classification and natural language inference across
    various categories including contradiction, entailment, and neutral
    classification. The model's architecture employs different
    classification heads on top of the same transformer, making it
    versatile for multiple applications. Notably, ModernBERT benefits
    from NLI training data such as the 'label-nli' dataset, specifically
    designed to enhance zero-shot classification abilities.)

Please predict the accuracy that tasksource/ModernBERT-large-nli would
    achieve on stanfordnlp/snli. Provide your answer as a JSON object
    with two keys: 'prediction' (a float between 0 and 1) and 'reason' (a
     brief explanation of your reasoning).
```

