# OpenReview forum: "ArtifactLinker: Linking Scientific Artifacts for Automatic SOTA Discovery"
_ICLR.cc/2026/Conference — ICLR 2026 Conference Withdrawn Submission_

### Official Review · Reviewer_97c7 · 2025-10-25

**Soundness:** 4
**Presentation:** 3
**Contribution:** 1
**Rating:** 4
**Confidence:** 4

**Summary:**

The paper ARTIFACTLINKER frames Hugging Face as a bipartite artifact graph and defines “automatic SOTA discovery” as finding high-potential missing links, then validating them via an agent that executes model→dataset→metric pipelines. The authors release ARTIFACTBENCH, specify the SOTA objective, a δ-thresholded candidate filter, four prediction tasks, and one verification task. Results show simple GNNs and graph-augmented LLMs outperform metadata/random baselines.

The problem is crisp and the evaluation protocol is mostly sound. That said, several results suggest shortcutting and fragility in the system stack (see Weaknesses).

The paper is readable and structured, with a useful pipeline schematic (Figure 1) and compact result tables (Tables 1–4). However, verification details are scattered.

The novelty is largely at the system level (pipeline + benchmark), not in modeling/representation learning. Link prediction uses off-the-shelf GATv2 and prompt-wrapped TextGNN; verification is a ReAct-style agent with HF-specific checks. The dataset/benchmark contribution is helpful but incremental relative to existing artifact knowledge graphs (see, e.g., LinkedPapersWithCode).

**Strengths:**

1. Clear problem setup: automatic SOTA discovery framed as link prediction on an artifact graph.
2. Novel system idea: combining prediction and executable verification for reproducible discovery.
3. Empirical evaluation across four prediction tasks + verification task.
4. Public resource (ARTIFACTBENCH) could stimulate follow-up research.

**Weaknesses:**

1. Limited methodological novelty: uses standard GNNs and ReAct agents; innovation is mostly system-level.
2. Strong shortcut bias: node-degree baseline F1=87.2, nearly matching GNN (88.4); no temporal or degree-controlled split.
3. Fragile verification: success rises on “hard” cases (1.3→14.3%) but drops on “easy” ones (56→43%).
4. No scalability or compute analysis: verification throughput and cost per edge are unreported.
5. Only Hugging Face is used as test dataset; unclear generalization to other artifact ecosystems and knowledge graphs (e.g., LinkedPapersWithCode, ORKG, etc.).

**Questions:**

1. How sensitive are your discovery results to the δ threshold in Eq. (5)? Did you analyze how δ affects the precision, recall, and number of verified links?
2. Can you provide a detailed breakdown of verification failures by stage (e.g., dataset loading, model initialization, metric evaluation, runtime errors)?
3. Have you evaluated the models under temporal or degree-controlled splits to rule out popularity or structural shortcuts in the graph?
4. How do you ensure that the prediction models (GNNs and LLMs) are not simply exploiting structural shortcuts like node degree or community membership? Have you tested temporal or degree-controlled splits or examined feature importance to verify genuine generalization?

---

> ### Author Response · Authors · 2025-11-27
>
> # Uniqueness of using Huggingface artifacts
>
> A core insight of our work is that HuggingFace is **unique** and provides the missing infrastructure for scalable automatic research. In contrast to papers or GitHub repositories—where execution requires reconstructing environments, resolving dependencies, and interpreting undocumented interfaces—HuggingFace offers a standardized, executable, and interconnected ecosystem. Three properties make it uniquely suitable for automated discovery:
> (1) **Uniform model usage** across models, datasets, and evaluation tools, enabling reproducible benchmarking; and
> (3) **Rich community-maintained metadata and versioning**, capturing real research dynamics.
> These features allow us building systems to reliably execute, validate, and compare hypotheses at scale—capabilities that are currently not feasible in any other research repository.
>
> # Main contribution
> Building on the Huggingface ecosystem, our contributions are:
>
> (1) A large-scale artifact graph benchmark constructed from HuggingFace models, datasets, evaluations, and metadata.
>
> (2) A new problem formulation—casting automated research as link prediction and link verification over the artifact graph.
>
> (3) A practical inference framework, Predict-and-Verify, enabling scalable automatic discovery by first proposing promising links and then validating them through execution on HuggingFace.
>
> Because the primary contribution is the task and benchmark, we intentionally adopt standard ReAct and GNN methods to establish clear, interpretable baselines. We view more advanced modeling and agent strategies as exciting future directions enabled by our benchmark.
>
>
> # Short-cut bias analysis
> We provide experimental evidence to show there is no shortcut bias in our results ,but our GNN actually learn structural features:
>
> (1) For link prediction tasks, when we increase the negative sampling ratio and scale the size of the artifact graphs, the gap between baselines and our proposed GNN methods becomes larger. Such results indicate that we do not have heavy popularity or structural bias in our evaluation tasks.
>
> $\downarrow$ Part of Table 1
> | Models (pos:neg=1:1 on original dataset)   | F1   | Precision | Recall | Accuracy |
> | -------- | ---- | --------- | ------ | -------- |
> |  Metadata        |    0.52  |     0.59      |    0.47    |  0.57        |
> |  Node degree        |   0.87   |  0.92         |  0.83      |   0.88       |
> |  GATv2Conv | 0.88 |    0.96      |   0.82     | 0.89     |
>
>
>
> $\downarrow$ New settings with more negative sampling
> | Models (pos:neg=1:5 on original dataset)   | F1   | Precision | Recall | Accuracy |
> | -------- | ---- | --------- | ------ | -------- |
> |  Metadata        |    0.31  |     0.22      |    0.50    |  0.63        |
> |  Node degree        |   0.62   |  0.85         |  0.49      |   0.90       |
> |  GATv2Conv | 0.82 |    0.87      |   0.77     | 0.94     |
>
>
> $\downarrow$ New settings with more negative sampling and an extended dataset
> | Models (pos:neg=1:5 on extended dataset)   | F1   | Precision | Recall | Accuracy |
> | -------- | ---- | --------- | ------ | -------- |
> |  Metadata        |    0.24  |     0.17      |    0.36    |  0.61        |
> |  Node degree        |   0.28   |  0.26         |  0.31      |   0.73       |
> |  GATv2Conv | 0.69 |    0.78      |   0.62     | 0.91     |
>
>
> (2) We also conduct an ablation study to show that the prediction also relies on text-embedding information. Therefore, information besides structural information is learned during GNN training.
>
>
>
> | Settings | F1 | Precision    |  Recall   | Accuracy |
> | -------- | -------- | --- | --- | -------- |
> | GATv2Conv w/doc embedding         |    0.884      |  0.958   |  0.821   |    0.892      |
> | GATv2Conv w/o doc embedding     | 0.824     |  0.898   |  0.772   | 0.835     |
>
>
> (3) Furthermore, we explain the phenomenon of the close performance between the degree baseline and the GNN method with **degree-controlled evaluation**. The conclusion is that the close gap between baselines and GNN models is caused by the high performance of those high-degree datapoints.
>
>
>
> | Settings  | Degree (<5) F1 | Degree (>5 and <20) F1 |
> | --------- | -------------- | ---------------------- |
> | Download  | 0.5352         | 0.8130                 |
> | Metadata  | 0.7925         | 0.9793                 |
> | GATv2Conv | 0.8338         | 0.9767                 |

---

> > ### Author Response · Authors · 2025-11-27
> >
> > # Code verification results
> > We acknowledge the inconsistent gains across easy and hard settings and provide an explanation in Line 409–410. For simple cases (e.g., evaluating `sst-2` with `bert-base-uncased`), the solution requires only a few lines of code, and decomposing the task into multiple ReAct stages can introduce unnecessary branching, error accumulation, or execution delays—ultimately hurting performance. In contrast, for harder scenarios involving complex dependencies or multi-step setups, this multi-stage structure is beneficial: it allows the agent to reason, verify, and correct sequentially, whereas an end-to-end ReAct pipeline often fails due to compounded errors. Thus, the variability reflects when decomposition is actually needed rather than a limitation of the framework.
> >
> > # Scalability and compute analysis
> > Verification throughput is around 5 minutes per datapoint. For the cost, it is around \\$0.3 for each datapoint. For the verification, thorough is thorough since we need to launch Docker, wait for the dataset and model download. For LLM cases, it might take more than 20 minutes for downloading and inference. We set a hard limit for the maximum time for each step.
> >
> > The execution time and cost are huge. However, since we conduct a predict-and-verify two-stage framework, we are able to control the verification step under a certain cost for scalable real-world discovery.
> >
> > # Threshold analysis
> > The $\delta$ parameter included in Eq.5 indicates the overall number of candidates we select that participate in the real-world link verification stage. It is highly heuristic and highly dependent on the overall cost that you want to conduct for verification. Since it is targeted for the real-world SOTA discovery, we do not have an appropriate benchmark for that. The current verification benchmark results are based on edge-number reproduction. Therefore, there is no direct connection between $\delta$ and the reported precision/recall/F1 scores in the paper. With a higher $\delta$, we have a higher chance to discovering the verified SOTA in our experiments, but its cost increases correspondingly.
> >
> > # Failure modes of verification
> > Since **React-linker operates as a three-stage pipeline**—dataset loading → model initialization → metric evaluation—we report success rates both cumulatively and conditionally:
> >
> > | Success rate | Dataset stage | Model stage | Metric stage |
> > | ------------ | ------------- | ----------- | ------------ |
> > | React-linker | 78.7%         | 45.1%       | 31.6%        |
> >
> > * **78.7%** of all runs successfully load the dataset.
> > * **45.1%** of all runs (i.e., **57.3% of those that loaded the dataset**) successfully initialize the model.
> > * **31.6%** of all runs (i.e., **70.1% of those with successful model initialization**) reach the final metric computation stage.
> >
> > These results show that **model initialization** is the primary bottleneck, with the lowest conditional success rate among the three stages. The potential reason is that the model is too large or there are some special requirements for special model loading.

---

### Official Review · Reviewer_Cdz5 · 2025-10-26

**Soundness:** 3
**Presentation:** 3
**Contribution:** 2
**Rating:** 4
**Confidence:** 4

**Summary:**

The paper targets the problem of efficiently discovering SOTA models for a given benchmark by leveraging existing resources. The authors construct a model-benchmark graph and propose a method to predict the performance of models on benchmarks based on this graph structure. They collect a new benchmark to evaluate their approach and provide insights for future developments in this area.

**Strengths:**

1. The paper targets at an interesting problem of efficiently discovering SOTA models for a benchmark by using the existing resources.
2. The paper collected a new benchmark to examine its problem and built a systematic framework to validate the problem.
3. The paper provides valuable insights for future developments in this area.

**Weaknesses:**

1. The discussion on related works is not thorough enough, please check the references and provided necessary discussions. Essentially, the construction of model-benchmark graph and prediction over it have been studied in previous works, which weaken the novelty of the proposed work.
2. The paper appears as an engineering prototype, to strengthen its contribution, it is better to reveal some unique patterns from the proposed setting, such as what sort of models are more likely to perform well on certain tasks, etc.

[1] ADGym: Design Choices for Deep Anomaly Detection

[2] Structuring Benchmark into Knowledge Graphs to Assist Large Language Models in Retrieving and Designing Models

[3] Beimingwu: A Learnware Dock System

**Questions:**

1. Why should there be four evaluation tasks? Selecting SOTA method is more about ranking the existing models in the correct order, as the absolute performance may vary with benchmarks and hard to predict.
2. What happens if a totally new benchmark is added, i.e., no prior information is available for this benchmark? How does the proposed method perform in this scenario? Since this is an important practical case of finding SOTA.

---

> ### Author Response · Authors · 2025-11-27
>
> # Uniqueness of using Huggingface artifacts
>
> A core insight of our work is that HuggingFace is **unique** and provides the missing infrastructure for scalable automatic research. In contrast to papers or GitHub repositories—where execution requires reconstructing environments, resolving dependencies, and interpreting undocumented interfaces—HuggingFace offers a standardized, executable, and interconnected ecosystem. Three properties make it uniquely suitable for automated discovery:
> (1) **Uniform model usage** across models, datasets, and evaluation tools, enabling reproducible benchmarking; and
> (3) **Rich community-maintained metadata and versioning**, capturing real research dynamics.
> These features allow us building systems to reliably execute, validate, and compare hypotheses at scale—capabilities that are currently not feasible in any other research repository.
>
> # Main contribution
> Building on the Huggingface ecosystem, our contributions are:
>
> (1) A large-scale artifact graph benchmark constructed from HuggingFace models, datasets, evaluations, and metadata.
>
> (2) A new problem formulation—casting automated research as link prediction and link verification over the artifact graph.
>
> (3) A practical inference framework, Predict-and-Verify, enabling scalable automatic discovery by first proposing promising links and then validating them through execution on HuggingFace.
>
> Because the primary contribution is the task and benchmark, we intentionally adopt standard ReAct and GNN methods to establish clear, interpretable baselines. We view more advanced modeling and agent strategies as exciting future directions enabled by our benchmark.
>
>
> # Missing related works
> Thank you for pointing out the missing related work. We have added discussion of these works in the revised paper. We emphasize that prior efforts primarily construct model–benchmark graphs and use them for model augmentation, whereas our contribution is an executable system for automatic discovery—specifically designed to identify unseen edges in artifact graphs.
>
> # Insights from experiments
> Besides the importance of structural information and the strong performance of GNN-based methods noted in Line 362 and Line 371, our experiments reveal additional consistent patterns:
>
> (1) **Larger models tend to rank higher**.
> Both GNN- and LLM-based approaches recover the intuitive trend that larger models generally perform better. For example, in our case study (Line 464–465), LLaMA-3-70B-Instruct is ranked above its 7B counterparts.
>
> (2) **Model–dataset specialization emerges naturally**.
> Models pretrained or fine-tuned for specific domains (e.g., biomedical QA, multilingual tasks, NLI) are preferentially linked to datasets from the same domain, indicating that the artifact graph captures specialization without requiring explicit metadata.
>
> (3) **Executability strongly affects discoverability**.
> Models with well-documented usage, standardized evaluation pipelines, and high deployment frequency (downloads) form more connections and are more frequently rediscovered—showing that reproducibility and interface quality meaningfully shape automatic discovery outcomes.
>
>
> # Evaluation task design
> In real-world automatic research, selecting the “best model” is not solely a ranking problem. Before ranking by performance (attribute ranking), the system must first retrieve which models/datasets are viable (link prediction). This follows a standard retrieve-then-rank workflow used in IR and recommendation systems.
>
> Therefore, for both link-level and attribute-level evaluation, we include prediction and ranking tasks for each: prediction serves as the necessary subtask that defines the candidate set, while ranking measures fine-grained ordering among those candidates.
>
> # Pipeline for new benchmark
> This scenario is exactly what our benchmark is designed to evaluate. In Section 9 (Lines 463–471), we present a SQuAD-v2 auto-discovery case study, where SGPT-125M surprisingly reaches EM = 0.41.
>
> A newly introduced benchmark is treated inductively: at inference time, it has no observed incident edges. If the release includes evaluation results, we treat those as initial edges; otherwise, we only observe the benchmark description. The model must then predict and rank which existing models are worth trying. This reflects the real workflow—new dataset released → retrieve and rank promising model candidates → run top-ranked pairs to verify and discover new links (as discussed in Section 4).

---

### Official Review · Reviewer_n9MX · 2025-10-29

**Soundness:** 2
**Presentation:** 3
**Contribution:** 2
**Rating:** 2
**Confidence:** 3

**Summary:**

An interesting topic, automatic SOTA discovery, is proposed, which actually tries to filter and predict the performance of existing models on existing benchmarks.

The paper first converts HuggingFace data into a bipartite graph. It then uses a straightforward information aggregation mechanism to predict performance, testing two routines based on existing methods. An LLM finally verifies the predicted potential SOTAs and runs the codes to obtain real performance. The experiment removes SOTA edges and then recovers them.

**Strengths:**

The paper is well-written, carefully wrapped up to emphasize its potential contribution.

The topic has an intriguing perspective, and the whole logic flow is self-contained.

**Weaknesses:**

1. The technical contribution is limited. While some LLM/agent papers adopt a similar high-level approach, this work lacks topic-specific design—such as deeper analysis of benchmark characteristics, dataset distributions, or model architecture encoding. Section 4.2.1 is particularly basic, using only multi-turn aggregations on 1-hop neighbors. The graph construction from HuggingFace is straightforward, and the ReAct-based verification appears to be a simple add-on. Table 3 lists only base models from existing works, which fails to support the claim that "the graph-based prediction module in ARTIFACTLINKER is effective in prediction."

2. The entire work relies on HuggingFace data, described as containing "millions of high-quality artifacts," yet produces a graph with only 1,372 models and 308 benchmarks. This small, single-source dataset is unconvincing. Discovery based on such limited data will be narrow and biased, regardless of method sophistication.

3. Presentation and grammar issues:

a. Figure 1 shows edges between model-model and benchmark-benchmark nodes, contradicting the "bipartite" design (Line 151, page 3). These edges are not explained in the method section.

b. "we propose a novel two-stage framework: (1) prediction and (2) verification." → "we propose a novel framework with two stages: (1) prediction and (2) verification."

c. "Prior work has largely relied on static analyses" → "Prior works have largely relied on static analyses"

**Questions:**

Q1. For novelty, I think I can hardly change my mind during the rebuttal phase. I will check the comments from other reviewers. The authors can focus on other issues.

Q2. For the dataset, please explain the unmatched size mentioned above. Is it possible to extend the scope from Huggingface to other data sources (something similar to paperswithcode)?

Q3. Please fix the grammar issues.

---

> ### Author Response · Authors · 2025-11-27
>
> # Uniqueness of using Huggingface artifacts
>
> A core insight of our work is that HuggingFace is **unique** and provides the missing infrastructure for scalable automatic research. In contrast to papers or GitHub repositories—where execution requires reconstructing environments, resolving dependencies, and interpreting undocumented interfaces—HuggingFace offers a standardized, executable, and interconnected ecosystem. Three properties make it uniquely suitable for automated discovery:
> (1) **Uniform model usage** across models, datasets, and evaluation tools, enabling reproducible benchmarking; and
> (3) **Rich community-maintained metadata and versioning**, capturing real research dynamics.
> These features allow us building systems to reliably execute, validate, and compare hypotheses at scale—capabilities that are currently not feasible in any other research repository.
>
> # Main contribution
> Building on the Huggingface ecosystem, our contributions are:
>
> (1) A large-scale artifact graph benchmark constructed from HuggingFace models, datasets, evaluations, and metadata.
>
> (2) A new problem formulation—casting automated research as link prediction and link verification over the artifact graph.
>
> (3) A practical inference framework, Predict-and-Verify, enabling scalable automatic discovery by first proposing promising links and then validating them through execution on HuggingFace.
>
> Because the primary contribution is the task and benchmark, we intentionally adopt standard ReAct and GNN methods to establish clear, interpretable baselines. We view more advanced modeling and agent strategies as exciting future directions enabled by our benchmark.
>
>
> # Construction process of artifact graph
> We include only a small subset of Hugging Face artifact graphs because we apply strict filtering to all collected relations. Specifically, we perform three stages of filtering:
>
> 1. **Documentation filtering.** We remove models and datasets without clear READMEs, model cards, usage examples, or citations. Many artifacts on public platforms are personal uploads with no evidence of real-world utility.
>
> 2. **Deduplication.** Popular datasets often appear under thousands of nearly identical repository names. For example, searching for GSM8K yields 2,083 variants (`HongzheBi/gsm8k`, `rookshanks/gsm8k`, etc.) alongside the canonical `openai/gsm8k`. We retain only the authoritative version.
>
> 3. **Exact name match validation.** We discard artifacts whose names cannot be unambiguously linked. For instance, `Undi95/MXLewd-L2-20B` reports a WER score on `open-llm-leaderboard`, yet no dataset with that exact name exists—only related entries like `vllms-leaderboard` or `vntl-leaderboard`.
>
> Consequently, we filter the pool to include only artifacts with clear documentation, demonstrated utility, and unambiguous name matching. As a result, our collected artifact graph is intentionally small, prioritizing precision and reliability over coverage. We incrementally make it bigger and we points details to **[scaling the size of artifact graphs]** part of the rebuttal.

---

> > ### Author Response · Authors · 2025-11-27
> >
> > # Scaling the size of artifact graphs
> >
> > To demonstrate scalability, we expanded the graph using the same pipeline with lighter and more clever filtering. By optimizing the filtering algorithm and incorporating additional metadata, the artifact graph increased from 1680 nodes / 2067 edges to 4060 nodes / 6206 edges. This required no architectural changes—only loosening matching constraints—showing that the graph size can grow substantially with additional effort or broader inclusion criteria.
> >
> > We see our resource as a **dynamic benchmark** that will also continue to grow since new resources are continuously added. We would release the larger version of the dataset upon publication.
> >
> >
> > | Settings | #Model Node | # Dataset Node | #Edge | #Avg Model degree | #Avg Dataset Degree |
> > | -------- | ----------- | -------------- | ----- | ----------------- | ------------------- |
> > | Original | 1372        | 308            | 2067  | 1.51              | 6.71                |
> > | Extended (exact match)         |   3383          |      677          |   6206    |        1.83           |      9.17               |
> > | Full (fuzzy match)         |     10174        |  3680              |   22686    |       2.23            |   6.17                  |
> >
> >
> > We also show that extended datasets makes our performance gain of GNN methods more convincing compared with the orignal one.
> >
> > $\downarrow$ Part of Table 1 in the paper
> > | Models (pos:neg=1:1 on original dataset)   | F1   | Precision | Recall | Accuracy |
> > | -------- | ---- | --------- | ------ | -------- |
> > |  Metadata        |    0.52  |     0.59      |    0.47    |  0.57        |
> > |  Node degree        |   0.87   |  0.92         |  0.83      |   0.88       |
> > |  GATv2Conv | 0.88 |    0.96      |   0.82     | 0.89     |
> >
> >
> >
> > $\downarrow$ New settings with more negative sampling
> > | Models (pos:neg=1:5 on original dataset)   | F1   | Precision | Recall | Accuracy |
> > | -------- | ---- | --------- | ------ | -------- |
> > |  Metadata        |    0.31  |     0.22      |    0.50    |  0.63        |
> > |  Node degree        |   0.62   |  0.85         |  0.49      |   0.90       |
> > |  GATv2Conv | 0.82 |    0.87      |   0.77     | 0.94     |
> >
> >
> > $\downarrow$ New settings with more negative sampling and extended dataset
> > | Models (pos:neg=1:5 on extended dataset)   | F1   | Precision | Recall | Accuracy |
> > | -------- | ---- | --------- | ------ | -------- |
> > |  Metadata        |    0.24  |     0.17      |    0.36    |  0.61        |
> > |  Node degree        |   0.28   |  0.26         |  0.31      |   0.73       |
> > |  GATv2Conv | 0.69 |    0.78      |   0.62     | 0.91     |
> >
> >
> >
> > # Presentation and grammar issues
> > Thanks for pointing that our presentation and grammar issues. We fixed those issues in the revised version of our paper.

---

### Official Review · Reviewer_rnjC · 2025-11-01

**Soundness:** 2
**Presentation:** 3
**Contribution:** 2
**Rating:** 4
**Confidence:** 4

**Summary:**

The paper models Hugging Face as a bipartite artifact graph between models and benchmarks, where edges carry evaluation scores.
Authors proposes a two-stage framework: prediction (GNNs or graph-augmented LLMs rank candidate links) and verification (an agent executes code to reproduce scores). They also build ArtifactBENCH to evaluate link/attribute prediction and reproduction.

**Strengths:**

- The paper is easy to follow. The problem formulation is clear.
- Authors present a benchmark (ArtifactBench) from hugging face and a set of evaluation tasks based on the benchmark.
- Authors conduct extensive experiments using heuristics, GNNs and LLMs+graph method.

**Weaknesses:**

- The size of the dataset is relatively small and the graph is sparse. Many edges are concentrated around a few popular datasets (e.g., ImageNet, MMLU, as shown in Figure 4). This may limit the usefulness of the prediction tasks that are around the nodes with smaller degree.
- For link prediction task, degree-based baselines already achieve high F1 (87.2 vs. 88.4 for the best method), suggesting the task may be too easy and may not present meaningful discovery.
- (Minor) ReAct-Linker appears to be a handcrafted pipeline running ReAct three times for different stages derived to solve the benchmark. Improvements are not consistent across easy and hard settings.  The contribution of this part feels small, though the authors also do not claim it as a major one.
- (Minor) The formulation of experiments is to exclude edges in the graph and predict them as targets. This setup focuses on re-discovery, which differs from discovering new, unobserved edges in a real-world setting (e.g., applying a model to a dataset that actually achieves new SOTA). Moreover, real-world discovery can be biased, especially with modern LLMs that already achieve SOTA across many datasets. This bias may limit the benchmark’s practical relevance.

**Questions:**

- Do you plan to scale the dataset and mitigate the concentration of edges on a few popular datasets?
- Do you plan to evaluate the framework on real discovery (unseen edges)?

---

> ### Author Response · Authors · 2025-11-27
>
> # Construction process of artifact graph
> We include only a small subset of Hugging Face artifact graphs because we apply strict filtering to all collected relations. Specifically, we perform three stages of filtering:
>
>
> 1. **Documentation filtering.** We remove models and datasets without clear READMEs, model cards, usage examples, or citations. Many artifacts on public platforms are personal uploads with no evidence of real-world utility.
>
> 2. **Deduplication.** Popular datasets often appear under thousands of nearly identical repository names. For example, searching for GSM8K yields 2,083 variants (`HongzheBi/gsm8k`, `rookshanks/gsm8k`, etc.) alongside the canonical `openai/gsm8k`. We retain only the authoritative version.
>
> 3. **Exact name match validation.** We discard artifacts whose names cannot be unambiguously linked. For instance, `Undi95/MXLewd-L2-20B` reports a WER score on `open-llm-leaderboard`, yet no dataset with that exact name exists—only related entries like `vllms-leaderboard` or `vntl-leaderboard`.
>
> Consequently, we filter the pool to include only artifacts with clear documentation, demonstrated utility, and unambiguous name matching. As a result, our collected artifact graph is intentionally small, prioritizing precision and reliability over coverage. We incrementally make it bigger and we points details to **[scaling the size of artifact graphs]** part of the rebuttal.
>
>
> # Edges in artifact graph
>
> While benchmarks like ImageNet and MMLU form expected hubs, edge concentration is limited. The average dataset degree is 6.71, and the median is 2.00, indicating broadly distributed connectivity. The top 5 datasets contribute only 25.88% of edges, and 45.13% of datasets have degree ≥ 3, showing that many non-famous datasets participate in multiple links. Thus, the graph is sparse but not dominated by a few popular nodes, and learning is not restricted to high-degree hubs.
>
> # Scaling the size of artifact graphs
>
> To demonstrate scalability, we expanded the graph using the same pipeline with lighter and more clever filtering. By optimizing the filtering algorithm and incorporating additional metadata, the artifact graph increased from 1680 nodes / 2067 edges to 4060 nodes / 6206 edges. This required no architectural changes—only loosening matching constraints—showing that the graph size can grow substantially with additional effort or broader inclusion criteria.
>
> We see our resource as a **dynamic benchmark** that will also continue to grow since new resources are continuously added. We would release the larger version of the dataset upon publication.
>
>
>
> | Settings | #Model Node | # Dataset Node | #Edge | #Avg Model degree | #Avg Dataset Degree |
> | -------- | ----------- | -------------- | ----- | ----------------- | ------------------- |
> | Original | 1372        | 308            | 2067  | 1.51              | 6.71                |
> | Extended (exact match)         |   3383          |      677          |   6206    |        1.83           |      9.17               |
> | Full (fuzzy match)         |     10174        |  3680              |   22686    |       2.23            |   6.17                  |

---

> > ### Author Response · Authors · 2025-11-27
> >
> > # Link prediction task difficulty
> > The strong degree baseline on the original graph is expected due to small size of datasets. To test whether the task is genuinely easy, we increased difficulty in two ways: (1) harder negative sampling (pos:neg changes from 1:1 to 1:5), and (2) evaluating on the extended artifact graph (2067 → 6206 edges) mentioned in **[Scaling the size of artifact graphs]**. Under both settings, degree performance drops sharply—F1 dropping from 0.87 to 0.62 and 0.28 separately—while GATv2Conv remains substantially stronger, dropping slightly from original 0.88 to 0.82 and 0.69. This demonstrates that the task is non-trivial and that meaningful relational modeling is required beyond simple structural heuristics.
> >
> >
> > $\downarrow$ Part of Table 1 in the paper
> > | Models (pos:neg=1:1 on original dataset)   | F1   | Precision | Recall | Accuracy |
> > | -------- | ---- | --------- | ------ | -------- |
> > |  Metadata        |    0.52  |     0.59      |    0.47    |  0.57        |
> > |  Node degree        |   0.87   |  0.92         |  0.83      |   0.88       |
> > |  GATv2Conv | 0.88 |    0.96      |   0.82     | 0.89     |
> >
> >
> >
> > $\downarrow$ New settings with more negative sampling
> > | Models (pos:neg=1:5 on original dataset)   | F1   | Precision | Recall | Accuracy |
> > | -------- | ---- | --------- | ------ | -------- |
> > |  Metadata        |    0.31  |     0.22      |    0.50    |  0.63        |
> > |  Node degree        |   0.62   |  0.85         |  0.49      |   0.90       |
> > |  GATv2Conv | 0.82 |    0.87      |   0.77     | 0.94     |
> >
> >
> > $\downarrow$ New settings with more negative sampling and extended dataset
> > | Models (pos:neg=1:5 on extended dataset)   | F1   | Precision | Recall | Accuracy |
> > | -------- | ---- | --------- | ------ | -------- |
> > |  Metadata        |    0.24  |     0.17      |    0.36    |  0.61        |
> > |  Node degree        |   0.28   |  0.26         |  0.31      |   0.73       |
> > |  GATv2Conv | 0.69 |    0.78      |   0.62     | 0.91     |
> >
> >
> >
> > # ReAct-Linker motivation
> > ReAct-Linker is not intended as a major algorithmic contribution, but rather as an instantiation of how our framework can operationalize edges—i.e., execute, verify, and compose link functions through agentic interaction. We use a three-stage ReAct pipeline simply because it is a practical and transparent baseline for this purpose. Our goal is to demonstrate that the benchmark supports executable reasoning over graph-structured knowledge, not to claim ReAct-Linker as a standalone methodological advance. We will clarify this framing in the paper. Although the gains are modest and not uniformly distributed across difficulty levels, they illustrate how agentic linking can improve functional edge grounding.
> >
> > # Discovery on unseen edges
> > We agree that rediscovery cannot fully substitute for evaluating genuinely novel edges. Mask-edge prediction is used because it provides measurable supervision, enabling controlled comparison across models. However, the framework is designed to generalize to unseen-edge discovery—ReAct-Linker already supports proposing and verifying new links. As noted in Lines 462–471, we conduct a real-world case study where SDPT-125M discovers new links on SQuAD-v2 (EM 0.41). Such evaluations currently lack standardized metrics, depend on human or domain-expert verification, and therefore we remain it as qualitative case studies rather than benchmark scoring. We will make this limitation and rationale more explicit and pose it as a challenge for future work since it is beyond the scope of our paper.
> >
> > For potential biases, open-source modern LLMs are also part of the artifact graph, so discovering their strong performance—even on older tasks like NER or parsing—is not a bias but an accurate reflection of the current research landscape and remains informative.

---

### Author Response · Authors · 2025-11-27

# General response

We appreciate the reviewers’ efforts in evaluating our paper. Below, we summarize the key points reviewers raised—items marked with ** indicate issues for which we provide additional experiments or clarifications, while unmarked items reflect strengths acknowledged by them. "Action/Summary" includes the highly summarized rebuttal content for each reviewer.

| Category                   | Reviewer rnjC                                                | Reviewer n9MX                                                | Reviewer Cdz5                                                | Reviewer 97c7                                                | Action/Summary                                               |
| -------------------------- | ------------------------------------------------------------ | ------------------------------------------------------------ | ------------------------------------------------------------ | ------------------------------------------------------------ | ------------------------------------------------------------ |
| Presentation               | "The paper is easy to follow..."                             | "The paper is well-written,..."                              | \*\*"**The discussion on related works is not thorough enough...**" | "Clear problem setup..."                                     | `Summary`: 3 of 4 reviewers agree on the clarity of our paper writing. `Reviewer Cdz5 rebuttal`: We added more discussion of related work in our revised version. |
| Novelty                    | NA                                                           | "The topic has an intriguing perspective...",\*\*"**The technical contribution is limited...**" | "...provides valuable insights...", \*\*"**...appears as an engineering prototype...**" | "Novel system idea...", \*\*"**Limited methodological novelty...**" | `Summary`: 3 of 4 reviewers believe the method novelty is not enough but the idea is novel. `Reviewer rnjC, n9MX, Cdz5 rebuttal`: We emphasize our contribution includes: (1) building a benchmark; (2) considering auto-research as link prediction; (3) proposing a predict-and-verify two-stage scalable framework. |
| HuggingFace as data source | "...present a benchmark (ArtifactBench) from hugging face..." | \*\*"**The entire work relies on HuggingFace data,...**"     | NA                                                           | \*\*"**Only Hugging Face is used as test dataset...**"       | `Reviewer n9MX and 97c7 rebuttal`: We explain the uniqueness of HuggingFace for conducting auto-research as link prediction tasks. |
| Size of artifact graph  | \*\*"**The size of the dataset is relatively small...Many edges are concentrated around a few popular datasets**"     | \*\*"**This small, single-source dataset is unconvincing...**" | NA                                                           | NA                                                           | `Reviewer rnjC & n9MX rebuttal`: We first point out that super nodes with high degrees are rare, the median degree is only around 2. Then we clarify that the strict filtering strategy led to the smaller initial dataset. We have expanded it to x3 sizes and consider it as a dynamic benchmark expanding through time. |
| Details of code verification          | \*\*"**ReAct-Linker appears to be a handcrafted pipeline...**" | NA                                                           | NA                                                           | \*\*"**...verification throughput and cost per edge are unreported.**" | `Reviewer rnjC rebuttal`: We reclaim that the motivation of React-linker is to provide potential solutions for code verification. `Reviewer 97c7 rebuttal`: We provide failure modes, efficiency, and cost analysis of code verification. |
| Potential short-cut bias       | \*\*"**...degree-based baselines already achieve high F1...**" | NA                                                           | NA                                                           | \*\*"**Strong shortcut bias: node-degree baseline**"         | `Reviewer rnjC & 97c7 rebuttal`: We find that baseline–GNN performance is close because high-degree nodes dominate and are easier to predict. With increased graph size and harder task setting, our method clearly outperforms the baselines. |

Beyond the above, we also discuss and clarify other aspects that were proposed by reviewers, including (1) the pipeline for inference on unseen edges, (2) more insights into which models work well on which datasets,  (3) the motivation for evaluation task design, and (4) threshold analysis in our inference algorithm.

---

### Note · Authors · 2026-01-05

I have read and agree with the venue's withdrawal policy on behalf of myself and my co-authors.